# Compact Current Reference Circuits with Low Temperature Drift and High Compliance Voltage

**DOI:** 10.3390/s20154180

**Published:** 2020-07-28

**Authors:** Sara Pettinato, Andrea Orsini, Stefano Salvatori

**Affiliations:** Engineering Department, Università degli Studi Niccolò Cusano, via don Carlo Gnocchi 3, 00166 Rome, Italy; sara.pettinato@unicusano.it (S.P.); andrea.orsini@unicusano.it (A.O.)

**Keywords:** selectable gain amplifier, resistive-sensor, current divider, current reference

## Abstract

Highly accurate and stable current references are especially required for resistive-sensor conditioning. The solutions typically adopted in using resistors and op-amps/transistors display performance mainly limited by resistors accuracy and active components non-linearities. In this work, excellent characteristics of LT199x selectable gain amplifiers are exploited to precisely divide an input current. Supplied with a 100 µA reference IC, the divider is able to exactly source either a ~1 µA or a ~0.1 µA current. Moreover, the proposed solution allows to generate a different value for the output current by modifying only some connections without requiring the use of additional components. Experimental results show that the compliance voltage of the generator is close to the power supply limits, with an equivalent output resistance of about 100 GΩ, while the thermal coefficient is less than 10 ppm/°C between 10 and 40 °C. Circuit architecture also guarantees physical separation of current carrying electrodes from voltage sensing ones, thus simplifying front-end sensor-interface circuitry. Emulating a resistive-sensor in the 10 kΩ–100 MΩ range, an excellent linearity is found with a relative error within ±0.1% after a preliminary calibration procedure. Further advantage is that compliance voltage can be opposite in sign of that obtained with a passive component; therefore, the system is also suitable for conditioning active sensors.

## 1. Introduction

Transistor-based current sources and current mirrors represent fundamental solutions in analog Integrated Circuits (IC) design [1], as well as in discrete circuits, whenever a biasing current source or a reference current is needed [2]. Current sources applications range from biasing and stabilization of circuits to reference or linearizing systems for conditioning of resistive-sensors widely used in industrial applications to measure different quantities (temperature, pressure, strain, and gas concentration, to name a few), as well as to set a test condition or simply regulate a signal for actuation. In these contexts, different solutions are available on the market [3]. Current generators capable of providing stable currents in the order of nA are also required for instrument calibration (i.e., picoammeters) and for materials characterization, both in experiments and in production tests [4]. As an example, in the widely studied field of nano-photonics and quantum photonics, it is of fundamental importance to have a stable low-level current source to precisely characterize the electroluminescence of fabricated devices [5,6].

Regarding resistive sensors, the basic signal conditioning circuit for allowing a measurable output voltage is the Wheatstone Bridge (WB), the output of which may in turn be converted into time intervals by means of time width modulators or into frequency by means of oscillators. In the literature, there are several low-cost solutions used to convert the sensor resistance variation into a time interval, such as Resistance to Period Converter (RPC) or Resistance to Pulsewidth Modulation (RPM) circuits [7,8,9,10]. Although WB is widely used, it shows the inconvenient of nonlinear dependence from the sensor impedance. Several linearization techniques have been reported, as the Current Mode WB [11], switches driven integration/deintegration method [12], and the double differential potential subtractor [13]. The latter only requires a constant and stable current flowing through resistive-sensor, as well as resistance matching. This method is well suited for conditioning resistive-sensors when relatively high resistances are concerned, i.e., the estimated error for a sensor resistance range of 15 kΩ–1.1 MΩ may be lower than ±1% [14,15,16,17]. In all the above-mentioned cases, it appears essential to provide a stable current in the range of 0.1 µA–10 µA flowing through resistive sensor. It is therefore clear how particular resistive-sensors are and how much they need a dedicated front-end capable of measuring resistance variations over wide ranges. As described, several solutions for resistive-sensors interfacing have been proposed. To evaluate the performance of these solutions, in some cases resistive-sensors were used [18]; more often, experimental evaluations were conducted using commercial resistors to simulate the sensor over a wide resistance range [19,20,21,22].

Even if high-performance dedicated monolithic current source ICs are commercially available [23], for particular applications it is necessary to design a specific circuit able to meet some given requirements [24]. At the circuit board level, the simplest ways to build such current source is to apply a linear voltage ramp to a differentiation capacitor or to use precision both voltage references and resistor in order to increase the system accuracy and ensure it over time [25,26]. However, they are mainly realized starting from op-amps acting as Voltage-to-Current (V-I) converters (i.e., transconductance amplifiers) [27]. V-I converters accept a voltage as input to source or sink a current from a load. They would also act as a current-reference if the input is a voltage reference. Since V-I converters output is a current, they need a load to work and, within their voltage compliance value, they work properly independently on the load nature.

Op-amp based Howland current-pump solution [28] is the most widely used circuit for the implementation of accurate V-I converters. For instance, they have been proposed as fully-integrated solutions for piezoresistive and resistive-sensors supply [29], as well as for electrotactile stimulation in sensory substitution systems for blind people [30]. Moreover, many Electrical Impedance Tomography (EIT) applications apply Howland circuits as current sources [31,32,33]. Howland circuit topology being similar to that of a difference amplifier (DA), low-power, high-precision integrated DAs have been recommended for current sources implementation [34]. Although several commercially available devices integrate both low noise op-amps and precision thin film resistors (with a fairly low temperature coefficient and an extremely good matching ratio), DA-based solutions require at least an external precision resistor to establish the desired V-I conversion factor. Conversely, a compact single-chip solution based on commercially available Selectable-Gain Amplifiers (SGA) requiring no additional external component has been proposed for Howland circuit implementation [35]. Due to the SGA internal thin film resistors excellent matching ratio and the low temperature coefficient, the designed circuit shows outstanding performance in terms of linearity, output resistance, and temperature dependence, with the possibility of implementing high-precision current sources operating in the µA–mA range. Moreover, its effectiveness for biasing and conditioning of platinum resistor-based temperature sensors has been verified [36].

In this paper, exploiting the aforementioned excellent performance of SGA devices, the realization and characterization of a precision current divider are illustrated. Experimental results highlight that circuit accuracy is mainly limited to that of the REF200 precision current reference [23] used to supply the single-chip current divider. Moreover, with a thermal coefficient lower than 10 ppm/°C in the range 10–40 °C, the circuit represents a good choice as on-board calibration system for compact pA-meters [26,37,38,39]. In addition, the adopted circuit architecture both guarantees a voltage compliance close to power supply rails and the separation of current carrying electrodes of the sensor and the sensing point for voltage reading. The system, tested on the field by emulating a resistive sensor with a sample of commercial resistors, shows excellent linearity over more than four decades of load resistance values. Taking also into account its compactness, the proposed system would represent a valid solution as front-end electronics in sensor-interfacing to AD conversion and processing circuitry.

## 2. Circuit Description

In this section, design consideration as well as simulated characteristics of two different current references are illustrated. Describing the system requirements, the next section will finally show the experimental results obtained on a prototype based on the design choices indicated hereafter.

### 2.1. SGA-Based Current Divider Circuit Analysis

Exploiting the tightly-matched on-chip resistor ratios, selectable-gain amplifiers LT1991/5/6 (by Analog Devices) can be configured into precision current dividers by strapping their pins [40]. With absolute values of tens of kΩ, LT1991 [41] integrates resistor with 1/1, 1/3, and 1/9 ratios, whereas 1/9, 1/27, and 1/81 are available for the LT1996 [42]. The latter well adapts for the realization of current dividers down to a hundredth by proper parallel connection of feedback resistors. Figure 1 shows an example of current divider based on the LT1996, able to source or sink an output current of about 1 µA with an input signal generated by the precision 100 µA current-reference REF200 chip [23]. Indeed, LT1996 pins have been connected for a 450 kΩ/109 resistance between op-amp output and its inverting input. By means of the 450 kΩ resistance between non-inverting input and output, the circuit acts as high precision 1:109 current divider.

Due to a resistor matching within ±0.05% [42], as worst case, a ±0.2% of accuracy is expected for the 1:109 ratio. For the proposed circuit, the positive compliance voltage of the output current is limited to about *V_CC_* − 1.2 V, which is the common mode voltage limit of the op-amp inputs. Conversely, a negative compliance voltage of about −*V_EE_* + 8 V is obtained, due to both the minimum 3 V supply for the REF200 and the 5 V voltage drop on the 50 kΩ resistor (at pin 10 of LT1996). Due to the dual power supply, negative values of voltage compliance allow to source also an active load.

It is worth to observe that the circuit is well suited to the conditioning of a high-resistance sensor used as load. Indeed, decoupled from the sensor itself, the voltage amplitude on the load can be measured at the low-impedance op-amp output node having *V_L_* = *V_MON_* − *V_OFF_*, with *V_OFF_*, around 0.42 V, the *V_MON_* value measured short-circuiting the load during a preliminary calibration procedure.

Equivalent current source non-idealities are mainly related to the input bias *I_B_* and offset *I_OS_* currents, as well as the offset voltage *V_OS_* of the op-amp. By inspection of the schematic of Figure 1, we obtain:(1)$$I_{OUT}=\frac{I_{\mathrm{IN}}}{109}+\frac{V_{OS}}{450\times 10^{3}}+\frac{I_{B\left(-\right)}}{109}-I_{B\left(+\right)}=\frac{I_{\mathrm{IN}}}{109}+\frac{V_{OS}}{450\times 10^{3}}-\frac{108}{109}I_{B}-\frac{110}{109}I_{OS}$$
where *I_B(+)_* = *I_B_* + *I_OS_* and *I_B(−)_* = *I_B_* − *I_OS_* are the bias currents absorbed by non-inverting and inverting inputs, respectively.

Assuming for *I_B_*, *I_OS_* and *V_OS_* the values declared in [42], typical 2.7 nA and maximum 5.2 nA absolute errors are then estimated for *I_OUT_*. Hence, current divider circuit accuracy would be within ±0.55%. In addition, a thermal coefficient (TC) of about 68 ppm/°C is also evaluated by Equation (1) taking into account the 7.5 nA maximum value for the bias current in the −40/+85 °C range, i.e., 60 pA/°C, and the *ΔV_OS_*/*ΔT* = 1 µV/°C (corresponding to 2.2 pA/°C for *I_OUT_*) values reported for the LT1996 [42].

Equivalent output resistance *ΔV_L_*/*ΔI_OUT_* of implemented current source should be evaluated by means of op-amp parameter dependence on common mode voltage *V_CM_* amplitude. A change in *V_CM_* will alter the operating point of op-amp input-stage giving rise to a change at the output and reflected at the input in the form of an offset error *ΔV_OS_* = *ΔV_CM_* / CMRR, where CMRR is the common-mode rejection-ratio of the op-amp. For the circuit illustrated in Figure 1, neglecting the voltage drop at the 5.56 kΩ (pin 3) resistor connecting load to the op-amp non-inverting input, *V_CM_ ≈ V_L_*, hence:(2)$$R_{OUT}=\frac{\Delta V_{L}}{\Delta I_{OUT}}\approx CMRR\frac{\Delta V_{OS}}{\Delta I_{OUT}}∼CMRR\times 450k\Omega$$
where the last equality has been found differentiating Equation (1) Assuming for CMRR the typical value of 120 dB for the LT1996 chip, a 450 GΩ value for *R_OUT_* is estimated. Although the absolute tolerance of LT1996 internal resistors is fairly poor (±30%), *R_OUT_* would range between 300 GΩ and 600 GΩ, a remarkable high value for the proposed reference-current generator.

In order to get a better insight into system characteristics, a circuit simulation has been performed by means of the equivalent SPICE model for LT1996. In order to evaluate main performance of LT1996-based current divider, an ideal 100 µA constant current generator has been used as REF200. Simulation result reported in Figure 2 outlines excellent performance in terms of the output resistance with a value as high as 700 GΩ. From Equation (1), it is worth noting that this value corresponds to a CMRR around 124 dB, a quantity declared for LT1996 for a voltage gain of 81 which is actually the ratio of the resistors seen at op-amp output.

A sink current is simply obtained by reversing the REF200 connections indicated in Figure 1 and obviously changing its supply voltage to *V_CC_*. In this case both the accuracy and the equivalent output resistance assume values equal to those estimated by previous analysis and performed by means of Equation (1) and simulations. Conversely, positive and negative compliance voltages assume almost complementary values to those aforementioned for the source current reference of Figure 1, i.e., −*V_EE_* + 1 V < *V_L_* < *V_CC_* − 8 V. In this case, too, it is possible to provide a current supply for an active load.

Simulating either source or sink output currents also allow to evaluate the amplitude of expected values of *I_OUT_* and *I_B_* (@ *V_L_* = 0 V). Indeed, from Equation (1), considering the two cases *I_OUT_* = *I_SOURCE_* (with *I_IN_* flowing as indicated in Figure 1) and *I_OUT_* = *I_SINK_* (with *I_IN_* sourcing pin 10 of LT1996), and neglecting voltage *V_OS_* and current *I_OS_* offset contributions we have:(3)$$I_{OUT}=\frac{I_{SOURCE}-I_{SINK}}{2}=917.431nA$$
(4)$$I_{B(+)}=\frac{I_{SOURCE}+I_{SINK}}{2}=2.48nA$$

The former is actually the expected *I_IN_*/109 value. Hence, Equation (3) allows the value of output current to be evaluated by eliminating the contribution due leakage currents of Equation (1) (and estimated with Equation (4)). This idea will be verified with experimental characterization carried out on the assembled prototype.

Finally, Table 1 summarizes main characteristics of the proposed circuit taking into account also REF200 features. It should be noted that other current-division values can be obtained simply by modifying the connections between LT1996 pins. For example, referring to the schematic of Figure 1, connecting pin 10 to 6 and using 8 as input, a 1:10 ratio is obtained.

### 2.2. 92 nA Current Reference Analysis

The presence of unused tightly-matched resistors at the op-amp non-inverting input of LT1996 suggests to achieve an additional division of the output current for the circuit of Figure 1. In particular, a 1:4 ratio is obtained by using resistors either at pins 1–2 or at pins 2–3, whereas a 1:10 factor is gained by means of the resistors connected at pin 1 and pin 3. In order to assure a current division independent of the *V_L_* load voltage, hence a high voltage compliance for the generator, used pins must work at the same *V_L_*. Figure 3 shows the schematic of the implemented circuit. By virtual connection of pins 1 and 3 of U2 performed by op-amp U3, output current is further divided by a factor of 10. Neglecting bias current and offset induced errors by both the two ICs, the load current is here equal to I*_IN_*/1090, therefore 91.74 nA as nominal value. It is worth noting that in this solution *V_L_* can be measured at U3 output (*V_MON_* pin in Figure 3). Unlike the above-mentioned circuit illustrated in Figure 1, in this case, the error is ideally represented only by the input-offset voltage of U3 and, again, can be evaluated by short-circuiting the load during a preliminary calibration procedure.

Although REF200 integrates two independent 100 µA references, in order to maintain the same accuracy, a double-pole, double-throw switch has been inserted allowing either sourcing or sinking current into the load by using the same generator. As aforementioned, in accordance with Equation (3), in this way, the (*I_SOURCE_ − I_SINK_*)/2 amplitude can be evaluated independently of the error induced by op-amps bias currents. This solution would reveal effective for calibration of a measuring instrument as underlined by the good accuracy found for the prototype as described in the next section.

As for the circuit shown in Figure 1, also in the new schematic the virtual connection that U3 makes on the output current divider should guarantee a constant output signal over a wide range of *V_L_* between the supply rails. In particular, for a source reference, S1 in position A, the highest value for *V_L_* coincides with the smallest common mode limit between U2 and U3. Conversely, as found for the aforesaid circuit, minimum value for *V_L_* is limited around *V_EE_* + 8 V. For a sink generator, S1 in position B, the voltage compliance will be in the range between −*V_LIM_* and *V_CC_* − 8 V, with −*V_LIM_* the highest negative common mode limit between U2 and U3. In both cases, for a purely passive load, the proposed circuit would assure a maximum load voltage fairly close to the supply rails, and dictated by common mode limits of either U2 or U3.

In order not to degrade the expected excellent performance of the current generator of Figure 1 highlighted in the previous section, U3 must be chosen within op-amp families with ultra-low offset voltage and bias current. In particular, by inspection of the circuit illustrated in Figure 3, the absolute leakage current on *I_OUT_* is now expressed as
(5)$$I_{leak}\approx I_{B(U3)}+I_{\mathrm{OS}(U3)}+0.1\times \left[I_{B(U2)}+I_{\mathrm{OS}(U2)}\right]+1.8\times 10^{-5}\times V_{\mathrm{OS}(U3)}+2.2\times 10^{-7}\times V_{\mathrm{OS}(U2)}$$
with terms having the same meaning as those of Equation (1). As expected, contributions introduced by LT1996 are now reduced by a factor of 10. On the contrary, the output current is directly unbalanced by the current *I_B(+)_* of U3, and its offset voltage gives a leakage quantity 81 times that induced by U2.

As shown by experimental results obtained for the realized prototype and illustrated afterwards, the dependence of op-amps input bias current cannot be neglected. However, as a rough estimation of TC, previous equation can be used to evaluate how thermal drift of op-amp offset voltages affect that of the output current:(6)$$TC\left(I_{OUT}\right)\approx \frac{1.8\times 10^{-5}\times TC\left[V_{\mathrm{OS}(U3)}\right]+2.2\times 10^{-7}\times TC\left[V_{\mathrm{OS}(U2)}\right]}{I_{\mathrm{OUT}\_\mathrm{NOM}}}$$
where *I_OUT_NOM_* = 91.74 nA represents the nominal value for *I_OUT_*. The contribution from U3 being 100 times larger than that of LT1996, Equation (6) stresses the need to use for U3 a device having a very low offset voltage drift, too.

For the proper choice of the device, circuit simulations have been performed for different op-amps and results are summarized in Table 2. The main characteristics of the op-amps used for U3 are shown in the first columns of the same table. Circuit simulations allowed to evaluate the output current amplitudes for either source or sink references, both for *V_L_* = 0 V. The *R_OUT_* values have been calculated as shown in Figure 2 evaluating the slope of the *I_OUT_*-*V_L_* characteristic outside any saturation condition. Finally, column *I_B_* shows the leakage current values, calculated with Equation (4), for *I_SOURCE_* and *I_SINK_* at *V_L_* = 0 V, while for all cases, Equation (3) gave the expected value of 91.743 nA.

Despite to superior performances obtained with LTC2054, it should be noted that the last three rows of Table 2 refer to low noise and precision devices that work in a more limited range of the supply voltage than the LT1996. Following the aforesaid requirements, by contrast OPA189 [43] was used for the final prototype. This device, with both common-mode voltage range and voltage supply comparable to that of the LT1996, should show the best performance thanks to the extremely high CMRR, as well as its very low bias current and the ultra-low absolute value and thermal coefficient of the offset voltage. Compared to the current divider alone, simulated output resistance of the circuit with OPA189 drops to 270 GΩ, a value however high enough to have an error in the order of only 0.04 nA for a 10 V compliance voltage value. It is worth observing that 270 GΩ corresponds to a CMRR of 134 dB which is actually the low-frequency value declared for OPA189.

## 3. Prototype Characterization

A current-reference circuit was assembled following the schematic illustrated in Figure 3. In order to reduce electromagnetic interference, the circuit was encapsulated in a metal box and a triaxial connector was used for the output with outer shield connected to line-earth. In Figure 4, a picture of the realized prototype is reported. At *V_L_* = 0 V, source and sink currents were (91.060 ± 0.002) nA and (−92.183 ± 0.002) nA, respectively, as measured by a Keithley 6517A electrometer. Hence, by Equation (3), an average value of (91.622 ± 0.004) nA is evaluated, in very good agreement with the ideal 91.743 nA, with an inaccuracy of −0.13% within that of REF200 (±0.25% typical). In addition, the absolute error of (0.562 ± 0.004) nA, estimated with Equation (4), is in perfect agreement with the amplitude of the current offset input for the OPA189 (up to ± 600 pA) although the other contributions expressed in Equation (5) cannot be ruled out. Disconnecting REF200 from pin 10 of LT1996, a (99.895 ± 0.09) µA value was measured from its output by means of the 6517A electrometer. Therefore, the evaluated 1:(1090.3 ± 0.9) factor underlines the excellent accuracy of on-chip resistor ratios of LT1996.

Output load regulation has been evaluated by measuring the output current amplitude as a function of the allowed range of output voltage *V_L_* generated by the source unit integrated in the Keithley 6517A. As expected, when REF200 biasing decreases below ~3 V, a sharp change in the output current value is observed. To highlight circuit performance outside any saturation condition, Figure 5 shows the output currents in a region around the 92 nA absolute values. By the data shown in the figure, an equivalent output resistance of about 100 GΩ, hence a 0.2 nA error in a wide range for *V_L_*, is evaluated. This value is almost three times lower than the estimated one reported in Table 2.

As expected by simulation and theoretical analysis, with the applied ±15.5 V supply voltage, a high voltage compliance is verified: −7.5 V < *V_L_* < 14.5 V and −14.5 V < *V_L_* < 7.3 V for the source (Figure 5a) and sink (Figure 5b) reference, respectively.

A climate chamber was used for temperature characterization of the circuit. The output current amplitude in the source configuration was acquired in the range 0–125 °C, although LT1996 performances are guaranteed up to only 85 °C. Experimental results are reported in Figure 6. A sharp decrease in the current value above 80 °C is clearly observed. However, as depicted in Figure 7, it is worth noting that in the range 0–60 °C the temperature drift, calculated by dividing the min-max current difference (156 pA) in the temperature range shown in the figure, is slightly lower than 30 ppm/°C and comparable to that of REF200. In addition, in the wide range around the ambient temperature (25 ± 15) °C, current is practically stable and no temperature drift is clearly observed.

To get an insight into the role that op-amps have on the temperature characteristics of the circuit, the | *I_OUT_* (T) − 91.622 Na|quantity has been calculated. Here, *I_OUT_* (T) represents data reported in Figure 6, whereas subtracting value is the above-mentioned current estimated by Equation (3) at ambient temperature. Hence, according to Equation (5), calculated values represent an estimate of leakage current due to both bias currents and offset voltages of op-amps. Plot of data, reported in Figure 8, shows that error contributions by op-amps is quite constant up to 40 °C. Conversely, an exponential behavior is clearly observed for *T* > 70 °C. In particular, current amplitude double every 9 °C (red dotted line) which is typical for reverse-biased pn-junction, hence attributed to diodes inserted for ESD protection of op-amp input stage. This increased bias current would be a significant problem for high temperature applications. However, the low thermal drift depicted in Figure 7 highlights excellent performance of the prototype in the relatively wide 0–60 °C range and underlines the good performance of chosen devices.

To evaluate LT1996 role on circuit performance, a further characterization was carried out on the same prototype but disconnecting U3. In this case, following the schematic depicted in Figure 1, the output current from pin 3 of LT1996 was measured. Experimental results are reported in Figure 9a in the range 0–125 °C. Here output current amplitude shows a sharp decrease for *T* > 85 °C, whereas it is almost constant, within ±300 pA, at lower temperatures. Hence, up to the maximum temperature of 85 °C declared for LT1996, a very low temperature coefficient (TC) of about 10 ppm/°C is found. From Equation (1), with data reported in [42] and neglecting the contribution due to resistors’ matching TC, the worst case for *I_OUT_* temperature dependence can be evaluated as:(7)$$\frac{\Delta I_{OUT}}{\Delta T}\approx 2.2\times 10^{-6}\left|\frac{\Delta V_{OS}}{\Delta T}\right|+\left|\frac{\Delta I_{B}}{\Delta T}\right|+\left|\frac{\Delta I_{OS}}{\Delta T}\right|\approx 40pA/{}^{\circ}C$$
i.e., a TC ≈ 44 ppm/°C. Taking into account also the REF200 thermal drift, as worst case a TC around 90 ppm/°C should be obtained. The lower value for TC experimentally estimated for the realized prototype highlights that a partial compensation between different contributions would exist.

The increase in op-amp bias current is also observed at the high temperature regime. Figure 9b shows the plot of data obtained subtracting the 916.6 nA average value measured at 25 °C to those of Figure 9a. In this case too, an exponential behaviour is clearly observed for *T* > 85 °C, with amplitude doubling every 9 °C (red dotted line). It is worth to note that *I_leak_* values are about one order of magnitude lower than those found with OPA189 inserted, highlighting again the good characteristics of the implemented LT1996-based current divider.

In order to better evaluate the temperature dependence degradation induced by OPA189, Figure 10 shows relative errors *r_ε_* for the two circuits calculated as:(8)$$r_{\epsilon }=\frac{I_{OUT}\;\left(T\right)-I_{OUT}\;\left(25\;{}^{\circ}C\right)}{I_{OUT}\;\left(25{}^{\circ}C\right)}$$
where *I_OUT_* (*T*) are the values acquired at temperature *T*, and *I_OUT_* (25 °C) the one acquired at ambient temperature.

For the 916 nA current source, open dots, a maximum error of about 0.3% up to 100 °C is found. Conversely, the 91 nA current source, where OPA189 is included, displays a relative error greater than 5% in the same temperature range. However, an error of about 2% is found in the temperature lower than 85 °C allowed for LT1996. It is worth to observe that this error is comparable to *I_OS_* + *I_B_* = ±2.6 nA declared for OPA189 [43] pointing out that most of the errors come from a change of the bias current at non-inverting input of U3 since the error induced by *ΔV_OS_*/*ΔT* of the chip is negligible.

To evaluate the long term stability of the output signal, a preliminary characterization has been performed. Raw data acquired continuously in 16 h are reported in Figure 11 together with a smoothed curve (red line). A ±30 pA peak-to-peak noise amplitude of raw data remains over the investigated time interval and the standard deviation is equal to 9.78 pA. The smoothed curve shows peak-peak fluctuations of about ±15 pA, too large to be attributed to laboratory ambient temperature fluctuations but likely depending on noise sources coupled to the circuitry.

To evaluate the system noise performance, the Keithley 6517A electrometer has been set to acquire 8192 subsequent samples in fast mode: integration time equal to 200 µs and sampling frequency around 79 Hz. Compared to data illustrated in Figure 11, where the instrument was set for an integration time of 20 ms, output current signal displays a higher noise content, with a root mean square value of about 0.2 nA. Amplitude spectral density of the output current was estimated by means of FFT algorithm implemented in MATLAB^®^ (R2019b). Results reported in Figure 12 (blue curve) state that an almost constant distribution is found in the investigated range with a value around 30 pA/√Hz. The same connection set-up was adopted to acquire a 92 nA signal generated by a Keithley 6221 precision current source, too. Results, reported in the same figure (red curve), highlight a less-noisy signal with values around 2 pA/√Hz for f < 10 Hz and as low as 0.6 pA/√Hz at higher frequencies. It is worth to observe that worst case estimation of noise signal induced by ICs as declared by manufactures [23,42,43] should be settle around 1 pA/√Hz level (see green dotted line in the Figure 12). Then, obtained results underline that the noisy-nature of the circuit could be tentatively attributed to a poor-shielding of the used circuit case.

Although a more detailed investigation of very long-term stability, as well as noise analysis performed by means of a spectrum analyzer, is required, experimental results shown here point out outstanding performance on the characterized prototype in terms of low thermal drift and high stability over time.

As already mentioned, the proposed circuit is suitable for conditioning resistive-sensors, which may have resistances ranging from a few kΩ to hundreds of MΩ. Finally, in this work an experimental test of the proposed circuit, for an output current of about 92 nA, was performed simulating a sensor with different commercial sample resistors. Each resistance value was measured with the Keithley 6517A high resistance meter. Conversely, an Agilent 34401A digital multimeter was used to acquire output voltage *V_MON_* values corresponding to the particular resistor connected at the input of current-reference prototype. Figure 13 summarizes obtained results for both *I_SOURCE_* (circles) and *I_SINK_* (rhombus symbols) currents. Axis on the right refers to the resistance values calculated by dividing the voltages by the 91.74 nA nominal value.

Experimental results demonstrate excellent performance in terms of linearity over more than four decades of resistance variation, from 10 kΩ to 122 MΩ. Best fit of experimental data (dotted line) gives slopes of (90.99 ± 0.02) nA and (92.02 ± 0.03) nA, in good agreement to the values preliminary measured by a Keithley 6517A at *V_L_* = 0.

The Figure 14a shows the relative error values of the resistors calculated using the nominal value of 91.743 nA, compared to the values measured with the 6517A Ohmmeter. It should be noted that the error is around −0.2% for voltages measured with *I_SOURCE_* (red circles) and −0.75% for those at *I_SINK_* (blue circles). The green squares, on the other hand, refer to the error calculated by considering the “average” voltage (*V_SOURCE_* + |*V_SINK_*|)/2, with which the offset voltage and bias current contributions of the U3 buffer are eliminated. In this case the error is about −0.6%, in agreement with the accuracy of the REF200 reference. It is worth noting that if the resistance value error is calculated taking into account the measured values of *I_SOURCE_* and *I_SINK_* at *V_L_* = 0 (Figure 14b), resistance values within ±0.1% error are obtained. This means that the system allows the resistive load to be accurately measured by a simple preliminary calibration operation with an ammeter connected in parallel to the load.

## 4. Conclusions

Current reference circuits are widely used in different applications, from biasing and stabilization of circuits, to resistive-sensors conditioning. In this work simple, high precision current references for grounded load have been described. The design and characterization of the realized prototype have been illustrated in terms of equivalent output resistance, thermal coefficient, as well as long-term stability. Main experimental results are summarized in Table 3. In particular, due to its excellent features, LT1996 selectable gain amplifier demonstrates particularly effective for the realization of a high-precision current divider. Two different ratios have been verified in the present work, but it is worth remarking that an appropriate choice of the LT1996 pin connections allows to obtain others *I_OUT_*/*I_IN_* values according to a specific requirement.

Supplied by a REF200 100 µA current reference, realized prototype exhibits good performance for either ~1 µA or ~0.1 µA source/sink current generator. Measured current values at ambient temperature are close to those expected by performed current division, with an absolute inaccuracy lower than 1%, mainly attributed to that of the REF200 IC. Voltage compliance extends at about 1 V of the supply rails (±20 V maximum) assuring good performance even for wide change of load-resistance. For a voltage supply of ±15.5 V, experimental results show that op-amps operate outside any saturation condition between −7.5 V and +14.5 V, and between −14.5 V and +7.5 V for source and sink reference, respectively. In this regard, the circuit also allows active load supply. Moreover, it is worth remarking that circuit architecture allows sensor-voltage sensing separated by current carrying sensor-electrodes, simplifying acquisition circuitry interfacing.

Realized circuit also demonstrates extremely good performance in terms of equivalent output resistance and low thermal coefficient (see Table 3). Experimental results show a relatively low noise amplitude of the generated current and a good stability has been verified up to 16 h. The noise performance of the system was also evaluated allowing to estimate an approximately constant distribution, with a value of about 30 pA/√Hz up to about 40 Hz to be mainly attributed to a poor shielding of the circuit.

The extremely low components count, up to three ICs and a few by-pass capacitors, means compactness and cost effectiveness of the proposed solution which would find effective application for resistive-sensor biasing as well as conditioning. In this regard, an experimental test was ultimately performed by simulating a sensor with a sample of commercial resistors. In the range from 10 kΩ to 122 MΩ, the system shows excellent performance in terms of linearity. In addition, the relative error values of the resistances have been calculated: errors are around −0.2% and −0.75% for the voltages measured with *I_SOURCE_* and *I_SINK_*, respectively. The error is around −0.6% when the average voltage value *(V_SOURCE_* + |*V_SINK_*|)/2 is considered, which compensates for the offset voltage and bias current of ICs. The error value is in good agreement with the accuracy of the REF200 reference. Moreover, thanks to a preliminary calibration, the system is appropriate to measure resistive loads accurately. In fact, the error reduces to ±0.1% if calculated considering the measured values of *I_SOURCE_* and *I_SINK_* at *V_L_ =* 0.

## Figures and Tables

**Figure 1 sensors-20-04180-f001:**
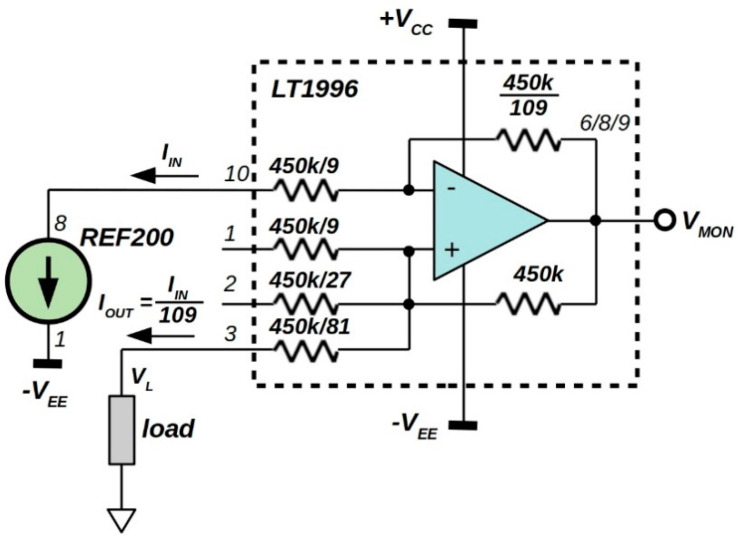
Precision current-divider based on an LT1996 selectable-gain amplifier. A REF200 current reference (by Texas Instruments) is used to have a source current of about 1 µA into the load.

**Figure 2 sensors-20-04180-f002:**
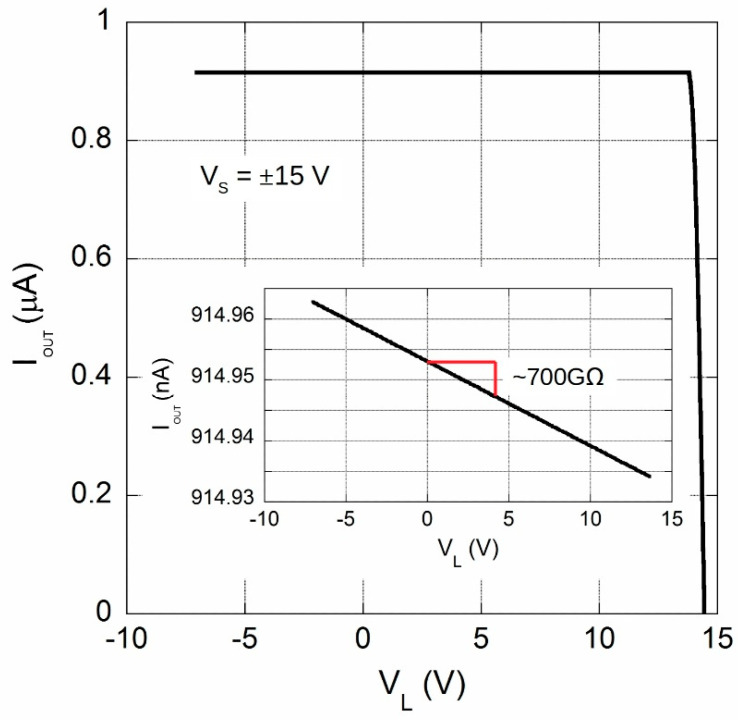
Simulated *I_OUT_* as a function of the load voltage *V_L_* for the circuit of Figure 1 using the SPICE model of LT1996 chip, whereas a 100 µA constant current was used of *I_IN_*. Simulation has been performed for the allowed *V_L_* > −7 V voltage range (see text). The inset reports the zoomed I-V characteristics in order to evaluate the equivalent output resistance.

**Figure 3 sensors-20-04180-f003:**
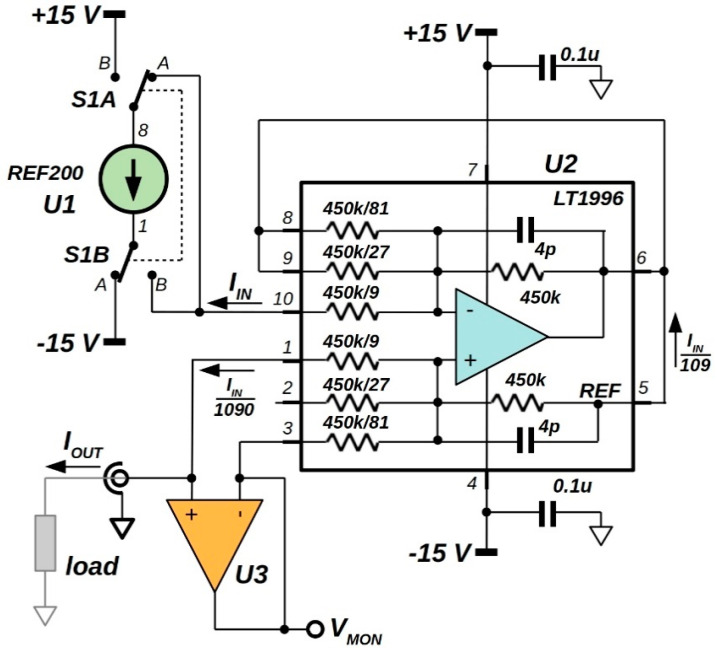
Schematic of the implemented current source. LT1996 acts as precision 1:109 current divider, whereas U3 allows a further 1:10 division of the output current. For a 100 µA source, a ~92 nA either source or sink current flows into the load by means of the S1 double-pole, double-throw switch.

**Figure 4 sensors-20-04180-f004:**
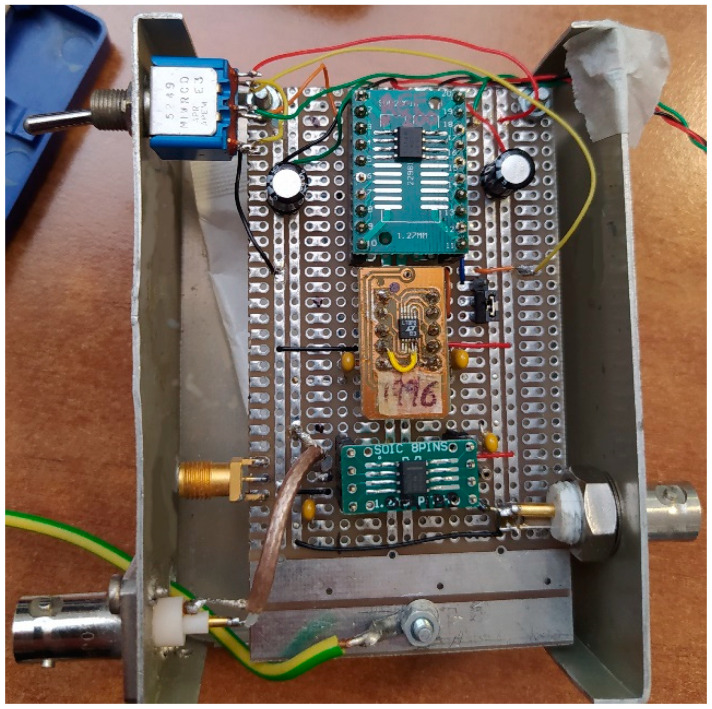
Picture of the prototype used for circuit characterization. The circuit was placed in a metal box. A triaxial cable with external shield connected to the earth-line was used for output connection.

**Figure 5 sensors-20-04180-f005:**
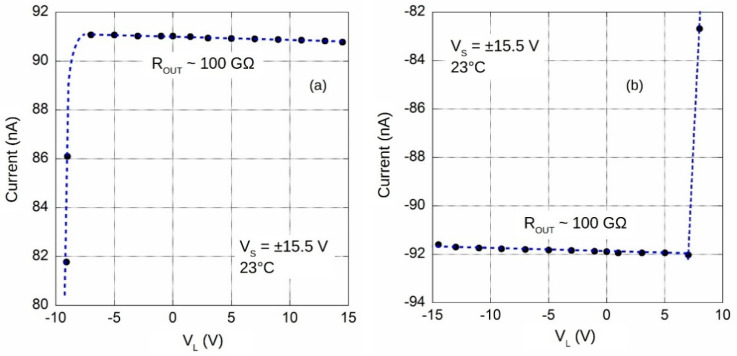
Output current as a function of the load voltage for both source (**a**) and sink (**b**) currents, with S1 of Figure 3 in A and B position, respectively.

**Figure 6 sensors-20-04180-f006:**
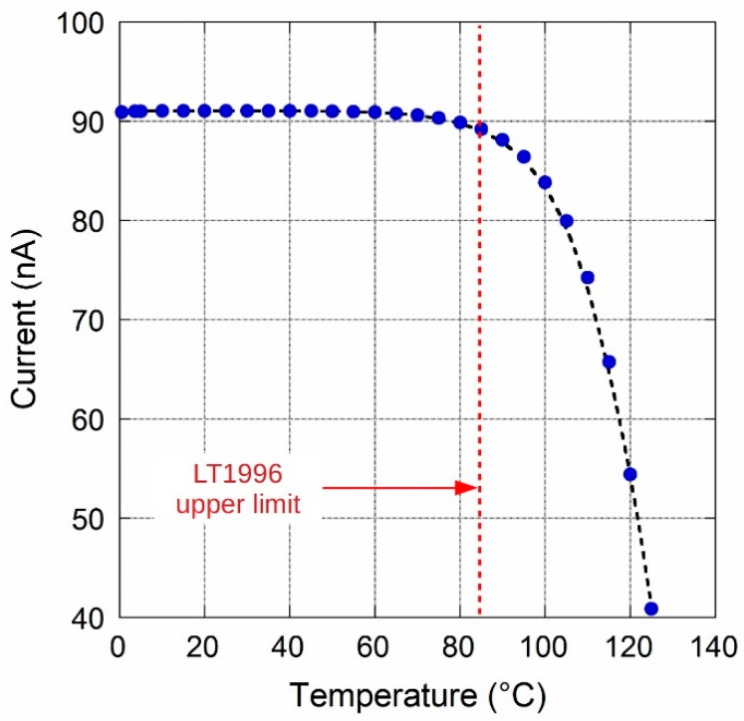
Circuit output source current *I_SOURCE_* values as a function of temperature.

**Figure 7 sensors-20-04180-f007:**
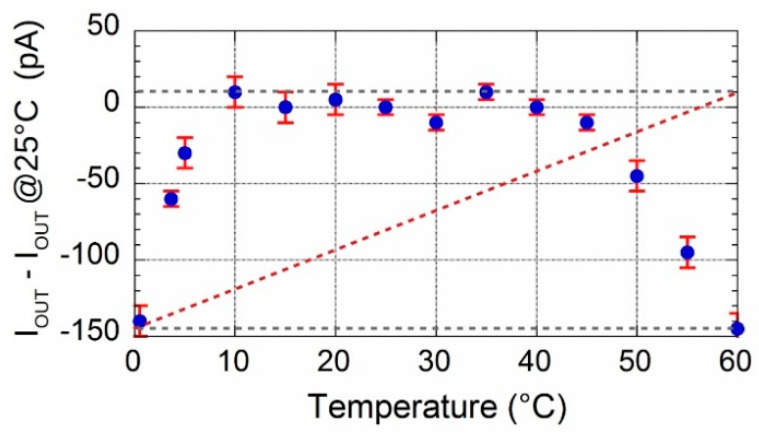
Currents have been calculated subtracting the value at 25 °C to evaluate the temperature drift in the 0–60 °C range.

**Figure 8 sensors-20-04180-f008:**
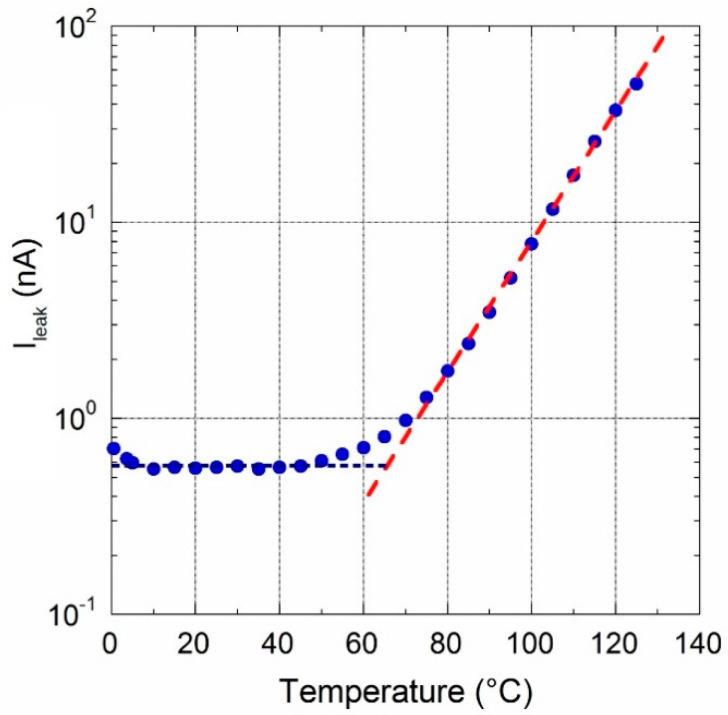
Current values calculated subtracting the 91.622 nA value estimated at ambient temperature to highlight leakage contribution due to reverse-biased op-amp input diodes (red dotted line).

**Figure 9 sensors-20-04180-f009:**
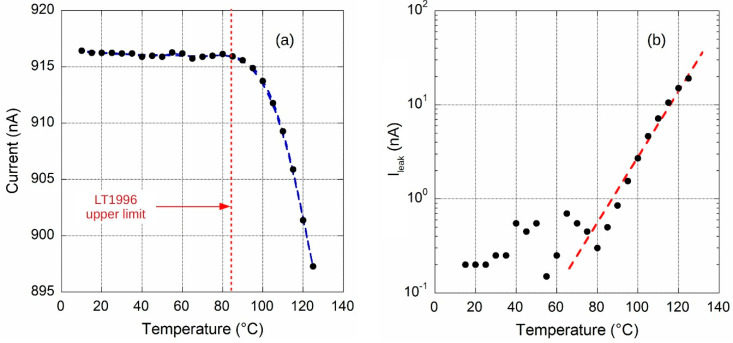
(**a**) Output of ~1 µA current reference circuit as a function of temperature. In (**b**), the value measured at 25 °C was subtracted to data reported in (**a**) to highlight leakage contribution due to reverse-biased op-amp input diodes (red dotted line).

**Figure 10 sensors-20-04180-f010:**
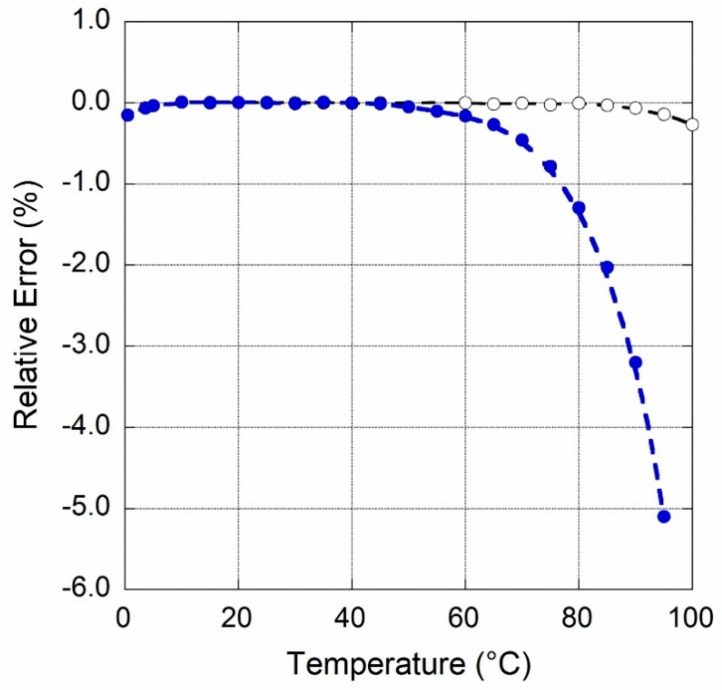
Relative error versus temperature for the circuits following the schematics of Figure 1 (open dots) and Figure 3 (full dots).

**Figure 11 sensors-20-04180-f011:**
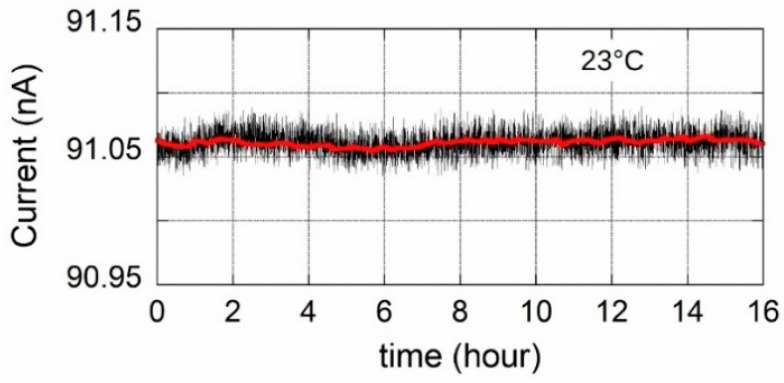
16 h of continuous acquisition of the output current.

**Figure 12 sensors-20-04180-f012:**
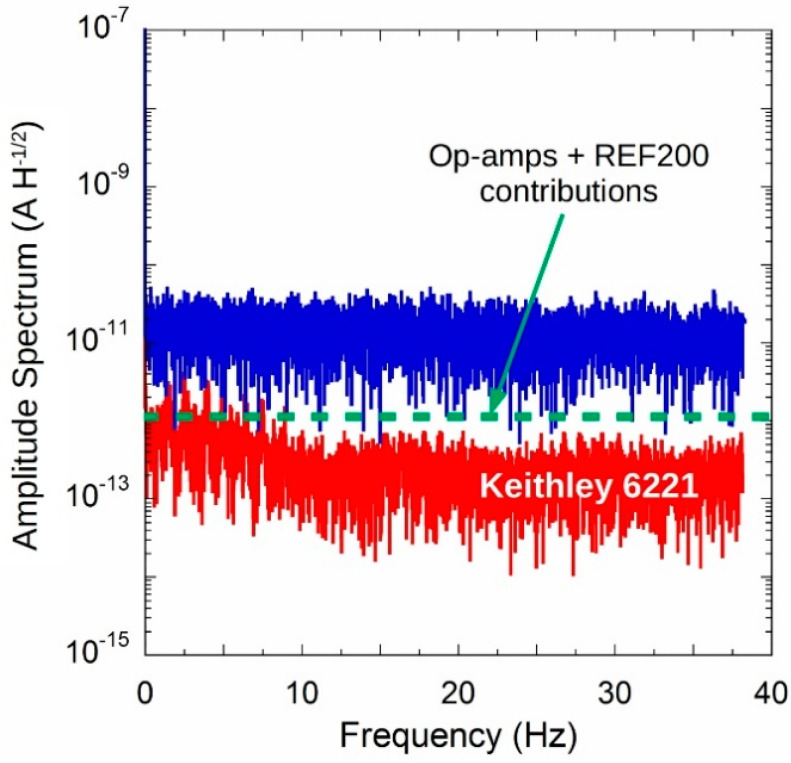
Amplitude spectrum of noise (blue curve) compared to that obtained in the same condition by a Keithley 6221 precision current source (red curve). Green dotted line represents the expected spectral noise level due to ICs used in the circuit.

**Figure 13 sensors-20-04180-f013:**
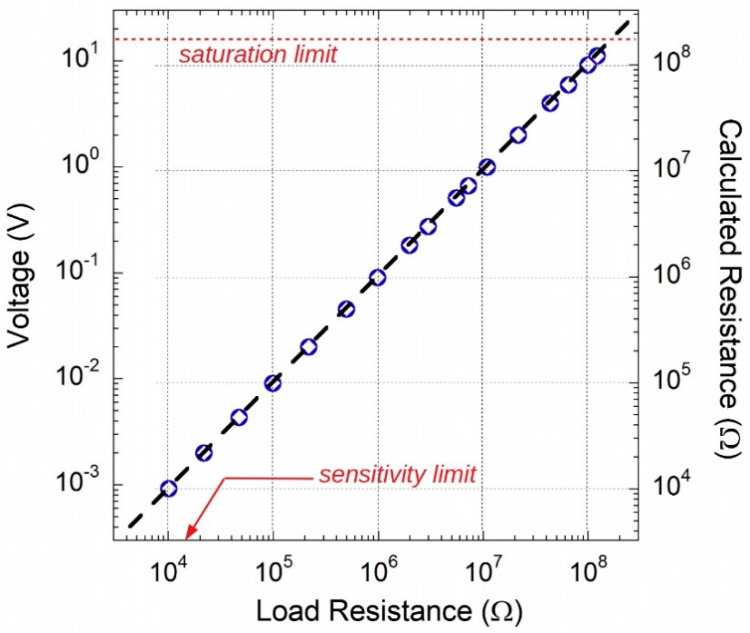
Proposed circuit exhibits an excellent linearity of the output voltage for load resistance values within the 10 kΩ–100 MΩ range.

**Figure 14 sensors-20-04180-f014:**
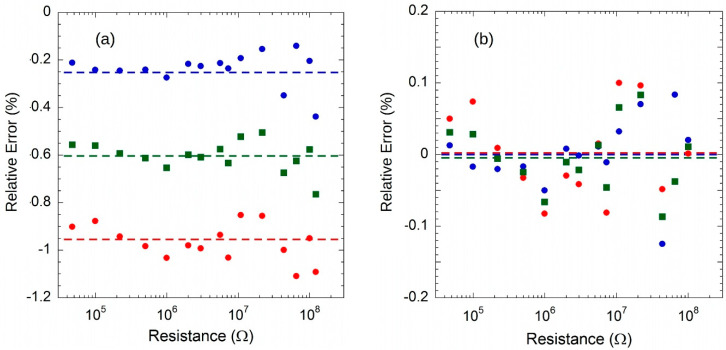
(**a**) Relative error of calculated resistance values for both *I_SOURCE_* (red circles), *I_SINK_* (blue circles) and “mean” (green squares) currents (see text). (**b**) Relative error if measured currents are considered (symbols have the same meaning of (**a**)). Dotted lines represent mean values of errors.

**Table 1 sensors-20-04180-t001:** Main characteristics evaluated for the circuit of Figure 1.

	Min	Typ	Max	Unit
**Current Value (nominal)**		917.4		nA
**Current Accuracy**		±0.54%	±1.55%	
**Temperature Drift**			95	ppm/°C
**Output Impedance**	500	700	900	GΩ
**Voltage Compliance (source)**	−*V_EE_* + 8 V		*V_CC_* − 1.2 V	
**Voltage Compliance (sink)**	−*V_EE_* + 1.2 V		*V_CC_* − 8 V	
**Supply Voltage**			±20	V

**Table 2 sensors-20-04180-t002:** Main characteristics and simulation results for some ICs chosen for U3.

U3	*I_B_* (nA)	*V_OS_* (µV)	*TC_OS_* (µV/°C)	CMRR (dB)	*I_OUT_SOURCE_* (nA)	*I_OUT_SINK_* (nA)	*R_OUT_* (GΩ)	*I_B_* (nA)
**OP07**	4	75	1.3	120	91.495	−91.991	38.7	−0.248
**OP27**	80	100	0.6	120	91.495	−91.991	5.7	−0.248
**OP1177**	3.8	61	2.2	126	93.488	−89.998	3.1	1.745
**OPA189**	0.3	3	0.005	168	91.430	−92.056	270	−0.313
**OP191**	65	500	1.1	90	120.01	−63.475	10.7	28.27
**LTC1022**	0.15	1000	3	92	91.495	−91.991	22	−0.248
**LTC2054**	0.003	10	0.1	130	91.496	−91.990	258	−0.247

**Table 3 sensors-20-04180-t003:** Measured electrical characteristics of realized prototype for the two circuits depicted in Figure 1 (1:109 ratio) and in Figure 3 (1:1090 ratio). Supply voltage ±15.5 V, *T* = 25 °C (unless otherwise specified).

	Min	Typ	Max	Unit
Supply Voltage (±V_S_)	8		20	V

Current Value (source)		91.06		nA
Current Value (sink)		−92.183		nA
Initial accuracy	−0.74		+0.48	%
Temperature drift				ppm/°C
10–40 °C			10	
0–70 °C		30		
Output Resistance		100		GΩ
Voltage Compliance (source)	−7.5		14.5	V
Voltage Compliance (sink)	−14.5		7.3	V
Noise Current (*f* < 3 Hz)		60		pA_P__-P_
		10		pA_rms_
Noise Current *BW* = 0.1 Hz to 40 Hz		30		pA/√Hz

Current Value (source)		916.6		nA
Current Value (sink)		−915.8		nA
Initial accuracy	−0.09		−0.2	%
Temperature drift 0–85 °C		10		ppm/°C
Voltage Compliance (source)	−7.5		14.5	V
Voltage Compliance (sink)	−14.5		7.5	V

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
