# Peer review of "Compact Current Reference Circuits with Low Temperature Drift and High Compliance Voltage"

_sensors, 2020, doi:10.3390/s20154180_

Round 1
Reviewer 1 Report
The authors provided very extensive and explanatory study on the proposed design of the compact current reference circuits with low temperature drift and high compliance voltage. The paper is well-written and provides interesting results, the figures are clear and understable. I have some minor recommendations:
- there is no need to use italic in text on page 2 for "i.e." and "external precision resistor", please correct,
- change the word "no-idealities" on page 3 to more suitable one, for example imperfections, non-idealities,
- considering the Figure 11 on page 12, change the graph for 24 h long one, if you have such measurement,
- maybe you can mention the estimate of the thermal noise contribution in conclusion with respect to the Figure 11, if you have such data.
Author Response
First of all, we would like to thank the reviewer for the precise and profound analysis of the manuscript: reviewers’ insightful comments have been welcomed, giving us the opportunity to revise the paper that has been improved, accordingly.
Please find hereafter a detailed lists of the answer and actions taken in order to comply with the reviewers’ comments.
The authors provided very extensive and explanatory study on the proposed design of the compact current reference circuits with low temperature drift and high compliance voltage. The paper is well-written and provides interesting results, the figures are clear and understable. I have some minor recommendations:
- there is no need to use italic in text on page 2 for "i.e." and "external precision resistor", please correct,
- change the word "no-idealities" on page 3 to more suitable one, for example imperfections, non-idealities,
Answer: We thank the reviewer for appreciating our work. We corrected all that reviewer mentioned. Changes have been highlighted in yellow in the revised version.
- considering the Figure 11 on page 12, change the graph for 24 h long one, if you have such measurement,
Answer: unfortunately, actually we cannot satisfy such a request because of the unavailability of a climate chamber. So, a 24 h long measurement should be performed in free air, at ambient temperature. Although the thermal coefficient is not so high, measurement would be affected by temperature changes in the lab during a day and data would not be comparable to what reported up to 16 h. We’ll perform such a characterization in the next future.
- maybe you can mention the estimate of the thermal noise contribution in conclusion with respect to the Figure 11, if you have such data.
Answer: We agree with the referee. We evaluated noise spectrum by FFT analysis of data acquired at a sampling rate of about 80 Hz. A new figure, was added and the numbering of figures was changed accordingly. The added comments to the figure have been reported in red color. In addition, thanks to the reviewer’s comment, we were able to add a new feature in the summary table reported in the conclusion section.

Reviewer 2 Report
This paper presents high-precision current references built with discrete components. Overall, the paper is well-organized and shows the detailed performance characteristics of the prototype. I have a couple of minor comments that should be addressed by the authors.
- As mentioned in the paper, showing Fig. 11 does not seem to be sufficient to demonstrate the long-term stability. I recommend the authors measure the frequency-domain spectrum of the output, which would not only show the long-term stability (this would be captured in the low-frequency region) but also show the noise performance.
- There are many typos or grammatically wrong sentences. Please proofread the manuscript and correct them.
Author Response
First of all, we would like to thank the reviewer for the precise and profound analysis of the manuscript: reviewers’ insightful comments have been welcomed, giving us the opportunity to revise the paper that has been improved, accordingly.
Please find hereafter a detailed lists of the answer and actions taken in order to comply with the reviewers’ comments.
This paper presents high-precision current references built with discrete components. Overall, the paper is well-organized and shows the detailed performance characteristics of the prototype. I have a couple of minor comments that should be addressed by the authors.
- As mentioned in the paper, showing Fig. 11 does not seem to be sufficient to demonstrate the long-term stability. I recommend the authors measure the frequency-domain spectrum of the output, which would not only show the long-term stability (this would be captured in the low-frequency region) but also show the noise performance.
Answer: We thank the reviewer for appreciating the work. We agree with the referee on the need for analysis in the frequency domain. We evaluated the noise spectrum by FFT analysis of data acquired at a sampling rate of about 80 Hz. As correctly stated by the reviewer, the analysis allowed us to evaluate the noise performance of the circuit. A new figure was then added and the numbering of figures was changed accordingly. The added comments to the figure have been reported in red color. In addition, we also added a new feature in the summary table reported in the conclusion section.
- There are many typos or grammatically wrong sentences. Please proofread the manuscript and correct them.
Answer: We corrected all typos and grammar errors, after a careful revision of the manuscript.
